# The Effect of a Novel Mica Nanoparticle, STB-MP, on an Alzheimer’s Disease Patient-Induced PSC-Derived Cortical Brain Organoid Model

**DOI:** 10.3390/nano13050893

**Published:** 2023-02-27

**Authors:** Nam Gyo Kim, Dong Ju Jung, Yeon-Kwon Jung, Kyung-Sun Kang

**Affiliations:** 1Adult Stem Cell Research Center, College of Veterinary Medicine, Seoul National University, Seoul 08826, Republic of Korea; 2Sol to B Co., Ltd., Gangnam-gu, Seoul 06242, Republic of Korea

**Keywords:** neurodegenerative disease modelling, high throughput screening, inflammation, autophagy

## Abstract

Alzheimer’s disease (AD) is one of the most well-known neurodegenerative diseases, with a substantial amount of advancements in the field of neuroscience and AD. Despite such progress, there has been no significant improvement in AD treatments. To improve in developing a research platform for AD treatment, AD patient-derived induced pluripotent stem cell (iPSC) was employed to generate cortical brain organoids, expressing AD phenotypes, with the accumulation of amyloid-beta (Aβ) and hyperphosphorylated tau (pTau). We have investigated the use of a medical grade mica nanoparticle, STB-MP, as a treatment to decrease the expression of AD’s major hallmarks. STB-MP treatment did not inhibit the expression of pTau; however, accumulated Aβ plaques were diminished in STB-MP treated AD organoids. STB-MP seemed to activate the autophagy pathway, by mTOR inhibition, and also decreased γ-secretase activity by decreasing pro-inflammatory cytokine levels. To sum up, the development of AD brain organoids successfully mimics AD phenotype expressions, and thus it could be used as a screening platform for novel AD treatment assessments.

## 1. Introduction

Alzheimer’s disease (AD) is one of the best-known neurodegenerative diseases, affecting more than 50 million individuals around the globe [1]. Cases in which AD is diagnosed in later life, beyond the age of 60, are referred to as sporadic AD (SAD), whereas in familial AD (FAD) patients develop AD symptoms at younger ages due to mutations in genes such as amyloid precursor protein (APP), presenilin-1 (PSEN1) and presenilin-2 (PSEN2) [2]. Mutations within the PSEN1 gene are known to cause the most severe forms of AD as an autosomal dominant neurodegenerative disorder [3]. Within the past decade, through vast amounts of research and many case studies, the neuronal deposition and subsequent toxicity of amyloid-beta (Aβ) plaques and hyperphosphorylated tau (pTau)-derived neurofibrillary tangles (NFTs) were established as the major hallmarks of AD. However, even though there has been substantial progress in the neuroscience of AD, there has been no significant improvement in AD treatments [4].

In the past several decades, the limitations of monolayer cell cultures have raised the need for an additional in vitro model system that could imitate the architecture of the human brain and its functions more precisely [1,5]. Furthermore, given the considerable differences between the human and mouse brain, phenotypes displayed in the mouse brain model are not able to fully represent those in the human brain, especially for studies of neurodegenerative disease [6,7]. To overcome these limitations, human induced pluripotent stem cells, which can self-organize, grow, and differentiate into specific linages, have been used to develop 3D human brain organoids [1,4]. Over the past several years, brain organoid models have been suggested to recapitulate normal brain development as well as pathological characteristics in neurodegenerative disease models [8,9,10]. Additionally, brain organoids can be generated from patient-derived induced pluripotent stem cells, allowing for personalized drug screening and the development of precision medicine [9].

Mica is a type of aluminosilicate mineral which is composed of various types of phyllosilicate minerals. Until recently, the presence of heavy metals within mica prohibited mica from being used as a medicine. However, STB-MP (Sol to B-Mica Powder) is a derivative of mica mineral, originating from South Korea, which has been processed in order to remove heavy metal residues and converted into nanoparticles, undergoing a dry cutting technique process, allowing STB-MP to be used as pharmaceutical graded compound [11]. The use of medical grade mica nanoparticles has recently been reported to have antitumor effects in colorectal and breast cancers when delivered as a nanoparticle [11]. In these studies, researchers demonstrated that STB-MP exerts immunostimulatory effects by upregulating the lysosomal and phagosome pathways [12], thereby activating macrophages. In the past decade, neuroinflammation within AD has not received much attention; however, in recent years, genetic and bioinformatic data from individual AD patients have provided the insight that inflammation participates in and aggravates AD pathology [13]. Therefore, we hypothesized that STB-MP might have a positive effect on reducing AD pathology.

In this study, we evaluated the efficacy of STB-MP in AD cortical brain organoids generated from patient-derived iPSCs. We hypothesized that STB-MP would be effective in reducing AD phenotypes by dampening neuroinflammation and activating autophagy. Overall, these findings indicate that this AD brain organoid model could be used as a screening platform to evaluate treatments for AD patients.

## 2. Materials and Methods

### 2.1. iPSC Culture

Two iPSC cell lines were used in this study. As described in our previous studies, KSCBi005, a human-induced pluripotent stem cell line from the National Stem Cell Bank of Korea (KSCB, Chung-Ju, Republic of Korea), and PSEN1-mutated patient-derived CS40iFAD-nxx (RRID:CVCL_YX94) iPSCs from the Cedars Sinai Medical Center iPSC Core Facility (CSMC, Los Angeles, CA, USA) were used to generate wild-type and AD brain organoids, respectively. These iPSCs were plated and cultured according to their manufacturers’ cell culture manuals. In short, KSCBi005 cells were cultured on fibronectin (Corning, Corning, NY, USA)-coated plates using Essential 8 medium (Gibco, Grand Island, NY, USA), and CS40iFAD-nxx cells were cultured on GFR-Matrigel (Corning, NY, USA)-coated plates in mTeSR^TM^ Plus medium (STEMCELL Technologies, Cambridge, MA, USA). Each cell line was passaged every 4–5 days, depending on the confluency of the cells, using ReLeSR^TM^ (STEMCELL Technologies, Cambridge, MA, USA).

### 2.2. Cortical Brain Organoid Culture

Cortical brain organoids were generated by following the protocol previously reported by Sloan et al. [14], with some alterations. Briefly, iPSCs were dissociated into single cells using Accutase. The dissociated cells were then counted and seeded into ultralow-attachment 96-well plates (Corning, NY, USA), with 9000 cells in each well with specific cell culture medium and 2.5 µM dorsomorphin, 10 µM SB431542 and 50 µM ROCK inhibitor. The medium was refreshed daily for 5 days. On Day 6, the formed EBs were then transferred into 60 mm culture dishes with neuronal differentiation medium consisting of Neurobasal-A, with 1 × GlutaMAX and 1 × B-27 Supplement without Vitamin A, 20 ng/mL bFGF and 20 ng/mL EGF. The medium was replaced daily for the first 10 days and then replaced every other day for the next 9 days. To further promote neuronal differentiation, neuronal differentiation medium supplemented with 20 ng/mL BDNF and 20 ng/mL NT3 was provided and replaced every other day from Day 25 until Day 42. From Day 43 onward, the neuronal differentiation medium was used without any growth factors and changed every other day.

### 2.3. STB-MP Characterization and Treatment

STB-MP is formulated with specific types of phyllosilicate minerals originating from South Korea. This specific combination of minerals was sliced down into nanoparticles via dry-cutting technique, slicing particles with the impact force of jet stream at the mach speed level. Atomic force microscopy (AFM) with an NX-PTR instrument (Park Systems, Suwon, Republic of Korea) was used to measure the size of the STB-MP nanoparticles. Powdered STB-MP was weighed and dissolved in PBS to the desired concentration and used at a 1:1000 dilution to reach the optimal concentration for treatment. Treatment was performed over a 2-week period with fresh medium changes every other day.

### 2.4. MTT Assay

WT iNSCs and PS1/2 KO iNSCs were seeded at 5 × 10^4^ cells/well into a 24-well culture plate in iNSC maintenance medium with ROCK inhibitor. After 24 h, STB-MP was applied at different concentrations for 48 h in triplicate. After washing each well with PBS, 500 μL of MTT solution (500 μg/mL) was added to each well and incubated at 37 °C for 4 h. Then, the same amount of DMSO was added to each well to dissolve the formazan crystals. The absorbance value at 570 nm was obtained immediately after adding DMSO using an Infinite200 PRO microplate reader (TECAN, Männedorf, Switzerland).

### 2.5. RNA Extraction and Reverse Transcription PCR

Five brain organoids from each group were washed with PBS and lysed in 1 mL of TRIzol (Invitrogen, Waltham, MA, USA) for RNA extraction. TRIzol treatment and total RNA extraction were performed following the manufacturers’ instructions. Complementary DNA synthesis was performed using Superscript-III First Strand KIT (Invitrogen, Waltham, MA, USA), following the manufacturers’ protocol. RT–PCR was then performed using SYBR Green PCR MIX (Applied Biosystems, Waltham, MA, USA), using a 7500 Real-Time PCR system (Thermo Fisher Scientific Inc., Waltham, MA, USA). All of the measurements were normalized to the level of the housekeeping gene GAPDH.

### 2.6. Western Blot Analysis

Five brain organoids in each group were washed with PBS and resuspended in 300 μL of Pro-Prep protein lysis buffer (Intron Biotechnology, Sung-Nam, Republic of Korea). Each protein sample was then resolved by SDS–PAGE and transferred onto a nitrocellulose membrane. The membranes were blocked for 1 h using a 3% bovine serum albumin (BSA) solution and then incubated overnight on an agitator with antibodies at a concentration recommended by the providers. Blots were washed and incubated with corresponding secondary antibodies at 1:2500 concentration for 1 h. Protein signals were detected using an ECL detection kit (GE Healthcare Life Science, Sung-Nam, Republic of Korea).

### 2.7. Histology and Immunofluorescence

At the designated timepoints, organoids were fixed in 4% paraformaldehyde for 1 h at room temperature. Each organoid was then washed 3 times with PBS, stored in 30% sucrose solution overnight at 4 °C, and embedded in cryo-solution with sucrose and gelatin for quick freezing using liquid nitrogen. Frozen organoids were then cryosectioned into 20 μm slices and stored at −20 °C. Sectioned organoid samples were washed with PBS, permeabilized and blocked using perm-block solution. The samples were then incubated overnight at 4 °C with the desired primary antibodies. The samples were then washed three times with PBS, incubated for 1 h at room temperature with secondary antibodies and stained with DAPI for 10 min. Each sectioned sample was mounted in DAKO fluorescence mounting medium (Agilent Pathology Solutions, Santa Clara, CA, USA). The images were taken using a confocal imaging microscope (Nikon, Tokyo, Japan).

### 2.8. ELISA

The protein levels of Aβ40 and 42 in RIPA buffer were measured using a human Aβ40 ELISA Kit (Thermo Fisher Scientific Inc., Waltham, MA, USA) and Aβ42 ELISA Kit (Thermo Fisher Scientific Inc., Waltham, MA, USA) according to the manufacturer’s instructions, and quantified using BCA analysis. Each measurement of Aβ40 and Aβ42 was taken from RIPA lysates of 5 pooled brain organoids, and triplicate measurements were performed for each group.

### 2.9. Statistical Analysis

All data analyses were conducted with Prism 9 software (GraphPad Software, San Diego, CA, USA), and statistical analyses were performed according to the number of groups, distribution, variance, and normality. Two-tailed Student’s *t* test analysis was employed for parametric datasets, and the level of significance was labeled as *** *p* < 0.001, ** *p* < 0.01 or * *p* < 0.05; no labels indicate a lack of significance. The number of biological replicates performed for each experiment is shown in each figure legend. Furthermore, all samples were chosen randomly for analysis.

## 3. Results

### 3.1. Generation and Characterization of Cortical Brain Organoids from AD Patient-Derived iPSCs

Two iPSC lines were used to model AD: CMC3 iPSCs were used as the wild-type group, and PSEN1 (PS1) iPSCs were used as the experimental group. Using these two cell lines, cortical brain organoids were generated following a previously established protocol with minor adjustments [14]. Briefly, cultured iPSCs were dissociated and seeded into ultralow-attachment 96-well plates for 5 days with two SMAD pathway inhibitors. On the sixth day of culture, neural spheroids were collected and transferred onto a culture dish for further differentiation into neural progenitors using Neural Medium. From Day 25 of culture, bFGF and EGF were replaced with BDNF and NT3 for further differentiation from neural progenitors into neurons until Day 42. Cortical brain organoids were then cultured for up to 8 to 14 weeks for analysis (Figure 1A). Ultralow-attachment 96-well plates were used to reduce the gap between organoids by controlling the size of each EB formed (Figure A1). On Day 43, sections of WT and AD cortical brain organoids showed positive expressions of the neural progenitor marker SOX2, the cell proliferation marker Ki-67, and the neuron markers TUJ1 and MAP2 (Figure 1B). These genes were also expressed at Day 84. Immunostaining images illustrated that AD patient-derived iPSCs show similar neurodevelopment compared to WT in developing cortical brain organoids. The quantified expression of specific markers, measured using ImageJ software, also illustrated that neuronal development and maturation in AD cortical brain organoids were similar to those in WT cortical brain organoids (Figure 1C). Together, the WT and AD cortical brain organoids generated from WT and PS1 iPSCs successfully recapitulate neurodevelopment in organoid cultures.

As described in the established cortical brain organoid generation protocol, organoid maturation begins at Day 43 [14]. Therefore, we extended the growth of organoids up to 12 weeks to examine the specific timepoint of AD phenotype expression. Tau expression was found at Day 43 in both WT and AD cortical brain organoids; however, phosphorylated tau expression was found only in the AD cortical brain organoids at Day 43 (Figure 1D). Additionally, Aβ was not expressed at all in either Day 43 or Day 84 WT cortical brain organoids, whereas its expression appeared in AD cortical brain organoids at Day 43, and Aβ aggregates began to form at Day 84 (Figure 1D,E). With evident phosphorylated tau formation and an increasing level of Aβ accumulation over time, AD cortical brain organoids successfully mimic AD phenotypes.

### 3.2. Characterization of STB-MP Nanoparticle Treatment and Toxicity

The size of STB-MP was measured using atomic force microscopy (AFM). STB-MP particles were then measured using the ImageJ program. A single STB-MP particle revealed a triangular prism shape with a base length of approximately 450 nm and a height of 15 nm (Figure 2A).

To obtain the effective STB-MP concentration range, the toxicity of the nanoparticles was measured. We first assessed 2D-induced neural stem cells (iNSCs). WT and PS1/2 KO iNSCs were generated from healthy and fAD patient-derived fibroblasts, as described in our previous study [15,16]. STB-MP was administered at concentrations ranging from 1 μg/mL to 1 g/mL, and toxicity was measured using the MTT assay. Cell survival rapidly decreased in both WT and PS1/2 KO iNSCs at a 1 g/mL concentration; therefore, a 100 μg/mL concentration was selected (Figure 2B).

### 3.3. STB-MP Treatment Does Not Affect Neurogenesis in Cortical Brain Organoids

STB-MP was administered to both WT and AD cortical brain organoids for 14 days, beginning at Day 84 (Figure 2C). After STB-MP treatment at a concentration of 100 μg/mL, there were no size differences between the treated and nontreated WT or PS1 organoids (Figure A2). Aside from the lack of size difference between organoids, there were no significant differences in terms of neuronal marker expression levels. The expression levels of neural markers such as Tuj1, MAP2 and NeuN were very similar in all groups of organoids. Additionally, immunostaining revealed that the number of cells expressing cleaved caspase 3, a known hallmark of cell apoptosis, was similar between the treated and nontreated groups in both WT and AD cortical brain organoids (Figure 2D). Taken together, these results indicate that the treatment of cortical brain organoids with 100 μg/mL STB-MPs does not affect neurogenesis.

**Figure 2 nanomaterials-13-00893-f002:**
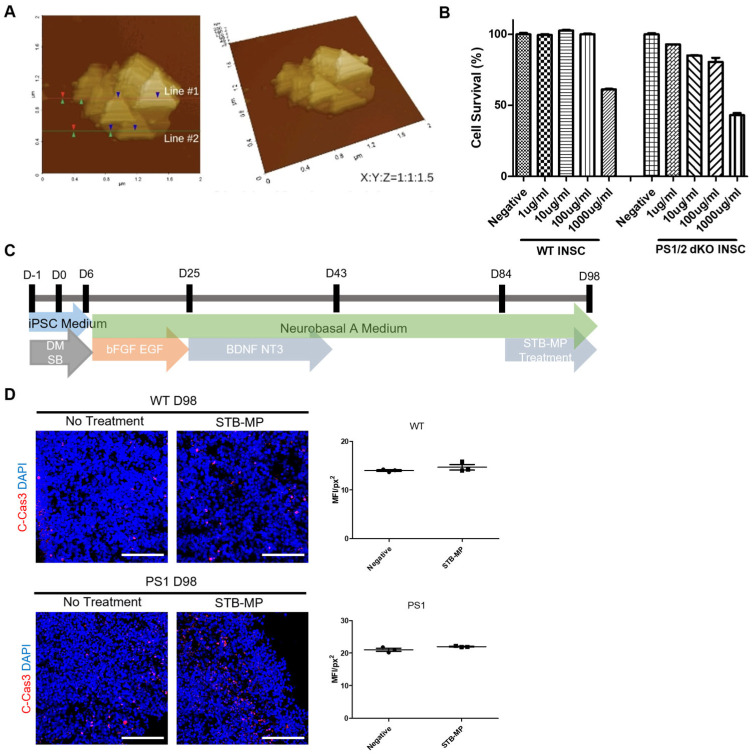
STB-MP nanoparticle characterization. (**A**) AFM image of STB-MP illustrating both the 2D and 3D shapes of the particles. (**B**) Toxicity testing of STB-MP in different cell lines. Serial dilutions from 1000 μg/mL to 1 μg/mL were used (n = 3). (**C**) Schematic image illustrating the STB-MP treatment schedule. (**D**) Representative images of immunostaining for the apoptotic marker c-Cas3 in WT and PS1 cortical organoids at Day 43 and Day 84 (n = 3). Scale bar 50 μm.

### 3.4. STB-MP Treatment Reduces Aβ Expression

STB-MP did not affect the neural development of AD cortical brain organoids, but did affect the degree of AD-related phenotypes. On Day 98 of culture, untreated AD cortical brain organoids showed an increase in Aβ expression. However, in the STB-MP-treated group, the level of Aβ expression was significantly decreased compared to that in the nontreated group. This decrease in Aβ expression was verified via Western blotting (Figure 3A,B) and ELISA (Figure 3C). The relative Aβ expression level decreased by approximately 48%, and the Aβ42/40 ratio also decreased by approximately 39% in the STB-MP group compared to the negative control group. On the other hand, the level of phosphorylated Tau expression did not show any significant changes as assessed by immunostaining or Western blotting (Figure 3A). These findings suggest that STB-MP penetrates cortical brain organoids and decreases Aβ expression.

### 3.5. STB-MP Treatment Mechanism of Action in AD Cortical Brain Organoids

Previous studies have suggested that mica nanoparticles (STB-MP) improve inflammatory responses [12]. STB-MP treatment of WT cortical brain organoids did not induce any changes in the levels of the proinflammatory cytokines TNFα, IL-1α and IL-1β or the anti-inflammatory cytokine TGFβ, compared to those in the untreated group. However, in AD cortical brain organoids, STB-MP treatment reduced the levels of proinflammatory cytokines (Figure 4A). Due to this reduction, the levels of innate immunity protein interferon-induced transmembrane protein 3 (IFITM3), known for increasing Aβ production by binding and upregulating the activity of γ-secretase [17], were also reduced in STB-MP-treated AD cortical brain organoids (Figure 4B–D).

As STB-MP reduces the activity of IFITM3, which then leads to the diminished activity of γ-secretase and reduced newly formed Aβ, the accumulated Aβ must be cleared via different mechanisms of action [18]. Therefore, we examined autophagy-related markers to determine whether STB-MP has an effect on the autophagy pathway. Both the STB-MP-treated and untreated PS1 cortical brain organoid groups were screened for autophagy-related protein expression. Both p62 expression and the LC3 II/I ratio were decreased, and Beclin 1 expression was increased in the STB-MP-treated group compared to the nontreated group. This implies that autophagy was activated by STB-MPs (Figure 4E). To determine its exact mechanism of action, AMPK and mTOR expression were also examined. There were significant changes in the p-AMPK/AMPK ratio and a decrease in the expression of p-mTOR, leading to a decrease in the p-mTOR/mTOR ratio in the STB-MP-treated group (Figure 4E). This indicates that STB-MPs activate autophagy by inhibiting mTOR.

## 4. Discussion

Along with the formation of neurofibrillary tangles driven by intraneural tau phosphorylation, one of the major hallmarks of AD is the formation and accumulation of Aβ in the brain, which results in the degeneration and loss of neurons [19,20]. AD leads to a further increase in inflammation, oxidative stress, the development of neural microvascular disease and ultimately dementia [21]. AD causes major cognitive defects and complications, yet there have not yet been safe or effective treatments developed for AD [22,23]. Moreover, although AD animal models or 2D cell culture models could be utilized to further investigate the effectiveness of drugs for AD treatment, these disease modeling systems do not fully recapitulate AD-specific phenotypes or particular indications observed in patients [24]. Previous studies have proposed the use of 3D organoids for modeling disease pathogenesis in multiple organs, such as the liver, brain, and intestine [25,26]. For instance, diverse types of brain organoids have been employed as tools to investigate the pathological development of neurodegenerative diseases such as Parkinson’s disease, microcephaly, and amyotrophic lateral sclerosis [27]. The main objective of our research was to confirm and verify that a human cortical brain organoid model generated from AD patient-derived iPSCs could successfully develop the necessary neuronal phenotypes and hallmarks of AD and be used as an AD drug efficacy screening platform.

In our work, we successfully generated AD cortical brain organoids using patient-derived iPSCs. These organoids express all of the necessary hallmarks of AD when cultured until a certain point. With this successful drug screening platform in place, the effect of mica nanoparticle STB-MP on AD cortical brain organoids was examined. With the advantage of its nanomolecular size, the administered STB-MP underwent passive uptake in AD cortical brain organoids. It did not damage the organoid or cause any neurodegenerative effects, and only exerted its therapeutic effect on its designated targets. STB-MP is known for its effect in reducing the inflammatory response [11], and STB-MP application reduced proinflammatory cytokines in AD cortical brain organoids. This led to a reduction in IFITM3 expression, thus indirectly reducing Aβ formation. Since STB-MP inhibits the generation of newly formed Aβ, we further examined different mechanisms that could eliminate already formed Aβ [28]. STB-MP tends to activate autophagy and aid in removing Aβ plaques by inhibiting mTOR. However, the detailed mechanism of action by which STB-MPs inhibit mTOR has not yet been fully examined.

This study has several issues and limitations that should be considered. First, as there were distinct changes in phenotypes in the STB-MP-treated group compared with the nontreated group, it was assumed that STB-MP is passively taken up into cortical brain organoids due to its nanosized structural advantages. However, the specific pathway of entry has not yet been studied. Second, STB-MP was administered for a 2-week period, and since the model does not have any excretion mechanism the long-term effect of STB-MP accumulation within the organoid has also not yet been studied. In vivo experiments focusing on the excretion mechanism of STB-MP will be critical to support the further application of STB-MP treatment [29]. In terms of in vivo experiments, there is no evidence that STB-MP can bypass the blood–brain barrier (BBB) and reach its therapeutic targets. Due to the size of STB-MP with nanomaterial characteristics, it is expected that STB-MP could cross the BBB; however, there needs to be a detailed study in order to carefully determine the correct method of administration for STB-MP to effectively cross the BBB and show a therapeutic effect on the patients. Therefore, a BBB penetration assay should also be considered [30,31].

## Figures and Tables

**Figure 1 nanomaterials-13-00893-f001:**
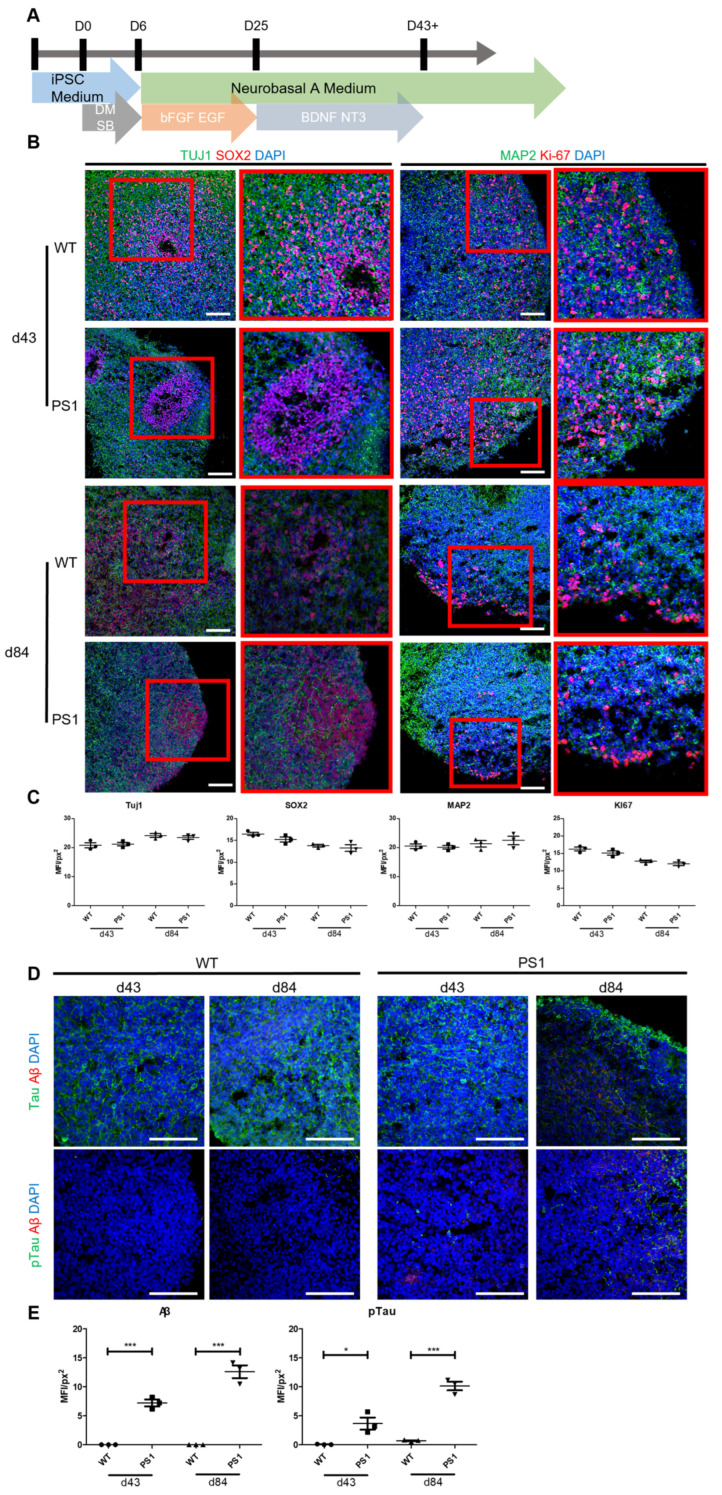
Generation and characterization of WT and PS1 hiPSC-derived cortical brain organoids. (**A**) Schematic image illustrating the timeline of hiPSC-derived cortical brain organoid development. (**B**,**C**) Representative images of immunostaining for cortical organoid markers Tuj1, SOX2, MAP2 and KI67 in sectioned WT and AD organoids at Day 43 and Day 84 (*n* = 3). Scale bar, 100 μm. (**D**,**E**) Representative images of immunostaining for AD phenotype-related markers Tau, pTau and Aβ in sectioned WT and AD organoids at Day 43 and Day 84 (n = 3). Scale bar, 50 μm. * *p* < 0.05, *** *p* <  0.001. The data are presented as the mean ± SD.

**Figure 3 nanomaterials-13-00893-f003:**
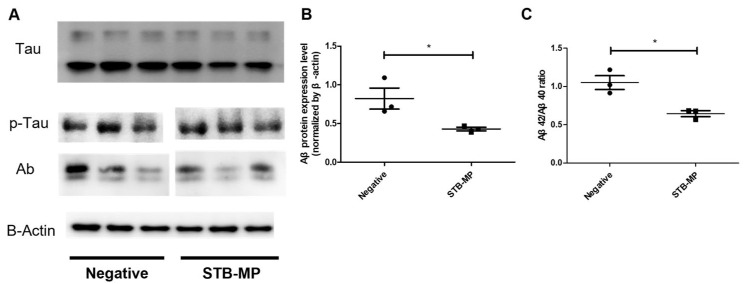
STB-MP nanoparticle diminishes the AD phenotype in PS1 cortical brain organoids. (**A**,**B**) Expression of the AD phenotype markers Tau, p-Tau and Aβ in WT and AD organoids at Day 98 was assessed by Western blotting analysis (n = 3, each data point represents 5 pooled organoids). (**C**) Lysates of WT and AD organoids were analyzed by ELISA at Day 98. The Aβ42/Aβ40 ratio in the RIPA fraction was measured by ELISA (n = 3, each data point represents 5 pooled organoids). * *p*  < 0.05. The data are presented as the mean ± SD.

**Figure 4 nanomaterials-13-00893-f004:**
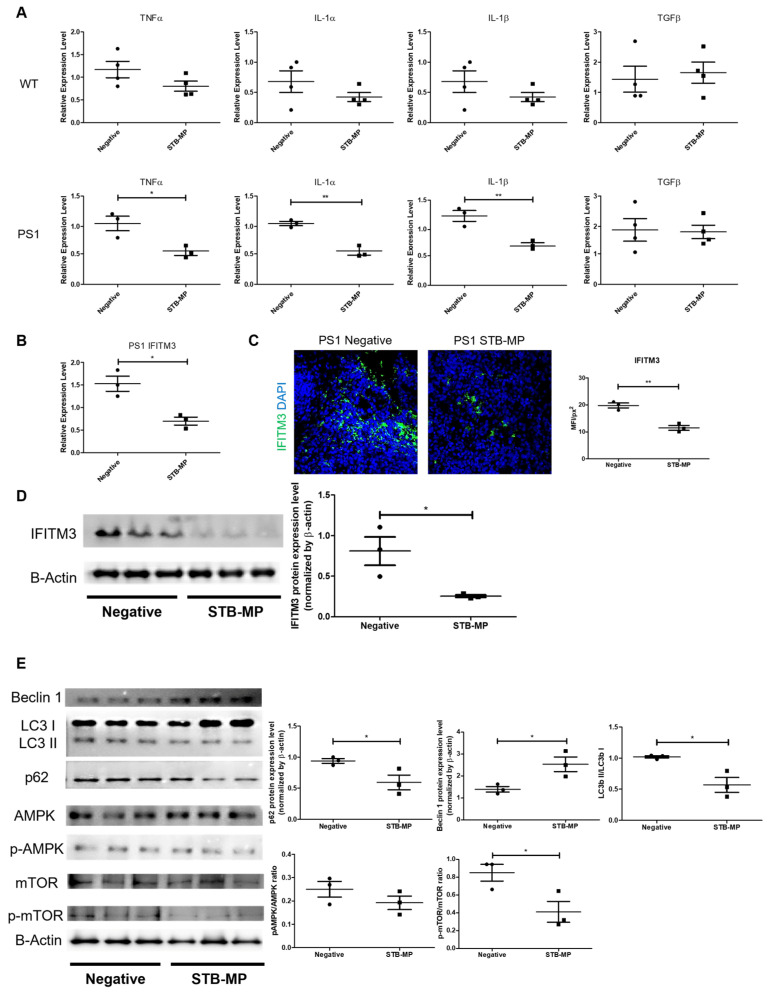
Inflammation and autophagy as mechanisms of action. (**A**) Quantitative real-time RT–PCR analysis of both pro- and anti-inflammatory cytokine expression was performed on treated and untreated AD organoids at Day 98. (**B**–**D**) IFITM3 expression analysis of the treated and untreated AD organoids at Day 98 by quantitative real-time RT–PCR, immunostaining, and Western blot analysis. (**E**) Western blot analysis and quantification of autophagy-related marker expression in the treated and untreated AD organoids at Day 98. n = 3 or 4, each data point represents 5 pooled organoids, * *p*  < 0.05, ** *p*  < 0.01. The data are presented as the mean ± SD.

## Data Availability

Data are available on request due to restrictions. The raw data presented in this study are available on request from the corresponding author.

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
