# Peer review of "The Effect of a Novel Mica Nanoparticle, STB-MP, on an Alzheimer’s Disease Patient-Induced PSC-Derived Cortical Brain Organoid Model"

_nanomaterials, 2023, doi:10.3390/nano13050893_

Round 1
Reviewer 1 Report
The manuscript of Kim et al. on the potential nanomaterial treatment of Alzheimer's disease contains noticeable data, but the current form needs revisions.
- First, the manuscript contains many pictures and graphics, but they are hard to evaluate in the presented miniaturized forms. Zooming results in very pixelated graphics that make them confusing. The reviewer recommends the authors include all figures in a higher resolution and a larger version in an SI file.
- The referee cannot understand the authors' intention with the repeated words of the title in the Keyword section. They need to substitute those repetitions.
- According to recent articles on the Nanomaterials homepage, the journal abbreviations need dots. The reviewer understands that the Instructions for Authors sometimes contain ambiguous recommendations, but visiting the relevant pages and articles of the target journal by the authors is not an excessive expectation. In addition, the abbreviated journal format is clearly visible at https://www.mdpi.com/authors/references.
Author Response
Q1. First, the manuscript contains many pictures and graphics, but they are hard to evaluate in the presented miniaturized forms. Zooming results in very pixelated graphics that make them confusing. The reviewer recommends the authors include all figures in a higher resolution and a larger version in an SI file.
Thank you so much for your recommendations. I have tried to convert the figure images to SI files, however, we were not able to find any converting tool that can be used for such conversion. On the other hand, I have converted all the images to TIFF file format and have replaced all of the figure images to TIFF in the newly uploaded manuscript. We would be more than happy to upload the original TIFF images to Nanomaterials.
The zoomed images (right) in Figure 1B are not just an enlarged image of the original image (left). We have taken both images using confocal microscope with x200 and x400 magnifications. We have added further explanation about the recording immunostaining images in the method section.
Q2. The referee cannot understand the authors' intention with the repeated words of the title in the Keyword section. They need to substitute those repetitions.
Thank you for your input. we misunderstood the use of keywords and thought that these words represent major points of the paper. we now have changed the keywords and now they do not contain any repetitions of the title. These words have been replaced with different keywords related to the paper to provide more information to the readers.
Q3. According to recent articles on the Nanomaterials homepage, the journal abbreviations need dots. The reviewer understands that the Instructions for Authors sometimes contain ambiguous recommendations, but visiting the relevant pages and articles of the target journal by the authors is not an excessive expectation. In addition, the abbreviated journal format is clearly visible at https://www.mdpi.com/authors/references.
We are very sorry, we misread the instructions on the referencing style. All the list of journals cited in the reference section are now in order with correct journal abbreviations and we have also added the ‘doi’ information on each journal.

Reviewer 2 Report
The authors present an interesting study combining nanoparticles and organoids as a new platform to get insight on AD treatments.
The research in of great interest, it is well developed, and results are convincing. There are, however, several questions to be faced.
1. STB-MP is used to describe mica nanoparticles, but what is the meaning of this acronym?
2. Point 2.3 in line 96 intends to describe STB-MP characteristics, but it is limited to the size determination. Are these particles purchased? If not, more information about their synthesis and characterization should be included.
3. The size of the nanoparticles is determined by AFM. Why have the authors chosen this method if there are more suitable ones such as cryoTEM? What is the size distribution of the nanoparticles?
4. Would the size of the nanoparticles be a drawback to cross the BBB?
Author Response
Q1. STB-MP is used to describe mica nanoparticles, but what is the meaning of this acronym?
STB-MP, which stands for Sol to B-Mica Powder, is a specific type of mica mineral formulated and sliced down into nanoparticle size. Thank you for mentioning this very important point and we have stated the full name of STB-MP when it was first mentioned in the Introduction section.
Q2. Point 2.3 in line 96 intends to describe STB-MP characteristics, but it is limited to the size determination. Are these particles purchased? If not, more information about their synthesis and characterization should be included.
STB-MP was purchased from Sol to B Co. Ltd., as mentioned in the Acknowledgement section. However, as the reviewer pointed out, we also believe that more information about the synthesis of STB-MP should be included in the paper.
STB-MP is formulated with specific types of phyllosilicate minerals originated from South Korea. This specific combination of minerals was sliced down into nanoparticles via dry-cutting technique; slicing particles with impact force of jet stream at the mach speed level. This technique is very unique as it allows to preserve the materials’ characteristics, Also, since it does not require a strong acidic solvent during slicing process, the level of toxicity is very low, thus STB-MP is currently used as a medicine.
The mentioned information above about the synthesis of STB-MP is now added in the Materials and Method section.
Q3. The size of the nanoparticles is determined by AFM. Why have the authors chosen this method if there are more suitable ones such as cryoTEM? What is the size distribution of the nanoparticles?
We acknowledge the reviewer for drawing our attention to this crucial point, which we consider to be of significant importance.
We specifically chose AFM for imaging STB-MP over other types of imaging technique due to following advantages. Firstly, AFM allows imaging samples without special sample preparation techniques and also, we were unsure whether flash-freezing in liquid nitrogen condition may alter the structure of STB-MP. Also, CryoTEM typically generates 2D images whereas AFM can generate high-resolution 3D images of nanomaterials. We wanted to show the exact shape and structure of STB-MP and believe that 3D images were more favorable than 2D images. For further analysis in the near future, we are also planning on utilizing CryoTEM for in vivo treatment samples, however this particular experiment is still at developmental stage and not yet been performed.
The size distribution of STB-MP is averaged around 100nm in length, according to Sol to B. However, regrettably, we are unable to provide graphical data on this matter due to confidentiality constraints.
Q4. Would the size of the nanoparticles be a drawback to cross the BBB?
We thank the reviewer for bringing to our attention the vital aspect of this experiment. In case of the size of nanoparticles crossing the BBB, we believe that STB-MP has the potential of crossing the BBB due to its small in size of around 100nm. However as mentioned in Discussion section, we have not yet completed in vivo experiments on STB-MP oral treatments. Therefore, we believe it is yet too early to state that STB-MP can cross the BBB.

Reviewer 3 Report
The article is original and very interesting.
The authors used Alzheimer’s disease patient-derived induced pluripotent stem cell (iPSC) o generate cortical brain organoids, expressing AD phenotypes; accumulation of amyloid-beta (Aβ) and hyperphosphorylated tau (pTau). Then they have investigated the use of a medical grade mica nanoparticle, STB-MP, as a treatment to decrease the expression of AD’s major hallmarks.
The topic is very relevant in experimental study of Alzheimer’s disease treatment. They have demonstrated that STB-MP inhibits the generation of newly formed Amiloid β and aid in removing Aβ plaques by inhibiting mTOR.
The conclusions consistent with the evidence and arguments presented and authors have accurately identified the limitations of the study.
The references appropriate, including some very relevant author’s previous experience in the field.
I suggest some minor corrections.
1 All the Figs have a Title on the top, then Title and Legend on the bottom. Delete the title from the top.
Line 82- you cite Pasca, but reference is Amin and Pasca. Then put 14 between backets [14]
Line 254 put 12 between backets [12]
1. In References section you may add the doi of the articles
Author Response
Thank you for your kind input. As you can see from the resubmitted paper, the titles were removed from the top of each figure. Also, the citations in line 82 and 254 have been corrected. Lastly, doi of the articles are added the reference section.

Round 2
Reviewer 1 Report
The quality of the figures has improved somewhat, and although many are still sandy, they are more readable.
The reviewer doesn't understand the author's answer about SI: what kind of conversion are they talking about?
The referee assumes they own the original images and figures. They need to create a Word/PowerPoint file, insert the images/figures (without compression) into the file then upload it as SI.